# Reversible modulation of interlayer stacking in 2D copper-organic frameworks for tailoring porosity and photocatalytic activity

Pei-Ye You[1], Kai-Ming Mo[1], Yu-Mei Wang[1], Qiang Gao [2], Xiao-Chun Lin[1], Jia-Tong Lin[1], Mo Xie[1], Rong-Jia Wei[1], Guo-Hong Ning [1] ✉ & Dan Li [1] ✉

The properties of two-dimensional covalent organic frameworks (2D COFs), including porosity, catalytic activity as well as electronic and optical properties, are greatly affected by their interlayer stacking structures. However, the precise control of their interlayer stacking mode, especially in a reversible fashion, is a long-standing and challenging pursuit. Herein, we prepare three 2D copper-organic frameworks, namely JNM-n ($n$ = 7, 8, and 9). Interestingly, the reversible interlayer sliding between eclipsed AA stacking (i.e., JNM-7-AA and JNM-8-AA) and staggered ABC stacking (i.e., JNM-7-ABC and JNM-8-ABC) can be achieved through environmental stimulation, which endows reversible encapsulation and release of lipase. Importantly, JNM-7-AA and JNM-8-AA exhibit a broader light absorption range, higher charge-separation efficiency, and higher photocatalytic activity for sensitizing $O_2$ to $^1O_2$ and $O_2^{\cdot-}$ than their ABC stacking isostructures. Consequently, JNM-8-AA deliver significantly enhanced photocatalytic activities for oxidative cross-coupling reactions compared to JNM-8-ABC and other reported homogeneous and heterogeneous catalysts.

Two-dimensional (2D) materials, including graphene, transition metal dichalcogenides, MXenes, and graphdiyne, have ignited great interest due to their intriguing optical and electronic properties, as well as quantum size effect. These properties are strongly affected by their interlayer interactions and stacking modes[1–3]. Thus, the precise modulation of interlayer stacking is a long-standing pursuit but is still hard to achieve. As a class of emerging 2D crystalline materials, 2D covalent organic frameworks (COFs) have been constructed from periodic and extended 2D monolayers that stack together through non-covalent interactions, including π – π interactions, electrostatic attractions, and van der Waals interactions[4–9]. Owing to their atomically precise and easily tunable structure, 2D COFs offer a promising platform for investigating how altering the stacking of layers affects their properties, such as porosity, crystallinity, chemical stability, and catalytic activities[10–12]. Therefore, many efforts have been made to modulate the

interlayer stacking of 2D COFs. For instance, the interlayer stacking modes of 2D COFs can be adjusted irreversibly via ligand engineering, including steric tuning and functional modification[13–17]. In addition, Zhao's group reported guest-triggered (e.g., solvent or gas molecules) reversible interlayer shifting in 2D COFs prepared from the identical monomers, but the resulting quasi-AA or -AB stacking structure cannot preserve in the absence of guests, leading to difficulty in comparing their properties[18,19]. So far, the preparation of structurally stable 2D COFs with different interlayer stacking modes but with the same components is rarely achieved.

Recently, we synthesized a series of copper cyclic trinuclear unit (CTU)-based 2D copper-organic frameworks (CuOFs) by integrating the chemistry of COFs and MOFs[10,20–22]. Due to the metallophilic attraction provided by CTUs, JNM-3 (JNM represented Jinan material) with staggered ABC stacking structure can irreversibly transfer to an

[1]College of Chemistry and Materials Science, and Guangdong Provincial Key Laboratory of Functional Supramolecular Coordination Materials and Applications, Jinan University, Guangzhou 510632, PR China. [2]CAS Key Lab of Low-Carbon Conversion Science and Engineering, Shanghai Advanced Research Institute, Chinese Academy of Sciences, Shanghai 201210, PR China. ✉e-mail: guohongning@jnu.edu.cn; danli@jnu.edu.cn

eclipsed AA stacking mode triggered by the addition of trifluoroacetic acid (TFA)[21]. Herein, we illustrated the reversible structure transformation between eclipsed AA and staggered ABC stacking, enabling reversible encapsulation and release of enzymes. With a pair of isostructures, the photocatalytic performance of CTU-based 2D CuOFs with different stacking structures was studied and compared. The imine condensation between Cu-CTU and three di-amine linkers produces three CuOFs, denoted to JNM-n (n = 7, 8, and 9) (Fig. 1). Interestingly, the reversible interlayer sliding between AA stacking (i.e., JNM-7-AA and JNM-8-AA) and ABC stacking (i.e., JNM-7-ABC and JNM-8-ABC) configuration can be achieved through environment cue simulation (i.e., solvent, acid, and heat) (Fig. 1). Importantly, their structures remain stable after the removal of solvent and acid, thus allowing the reversible encapsulation and release of lipase. Furthermore, JNM-7-AA and JNM-8-AA exhibit a broader light absorption range, higher charge-separation efficiency, and better photocatalytic activity for sensitizing $O_2$ to $^1O_2$ and $O_2^{\cdot-}$ compared to their ABC stacked isomers. Consequently, JNM-8-AA demonstrates remarkably higher photocatalytic activities for oxidative cross-coupling reactions than JNM-8-ABC. We can harness the respective advantages of MOFs and COFs via combination of coordination and dynamic covalent chemistry, allowing us to precisely control the interlayer stacking of 2D materials and understand the structure-property relationship.

## Results

### Synthesis and characterization

The synthesis of JNM-7-AA, JNM-8-AA, and JNM-9-ABC were conducted under solvothermal conditions (Fig. 1). Typically, a mixture of 1,2-dichlorobenzene (o-DCB), 1-butanol (n-BuOH), and 6 M trifluoroacetic acid (TFA) containing Cu-CTU and di-amine linkers (i.e., 4,4'-diaminobiphenyl (DABP), 4,4'-diamino-p-terphenyl (DATP) or 4,4'-diamino-p-quaterphenyl (DAQP)), was heated at 120 °C for 72 h. Afterward, pale-yellow or dark-brown crystalline powders were obtained with 70% – 97% yields (Fig. 1 and See Method).

As shown in Fig. 2a, the powder X-ray diffraction (PXRD) patterns of JNM-7-AA feature an intense peak at a low angle of 2.56° attributed to (100) reflection facet along with the minor peaks at 4.52° and 6.92° for (110) and (120) diffractions, respectively. To elucidate the crystal structures of JNMs, three possible configurations, including eclipsed stacking (AA) and staggered stacking (AB and ABC) modes, were simulated by the Materials Studio software. The experimental PXRD patterns of JNM-7-AA matched well with the simulated profiles of AA stacking mode (Fig. 2a and Supplementary Fig. 2). Pawley refinement of JNM-7-AA gave a hexagonal space group P6/m with unit cell parameters of a = b = 41.2260 Å and c = 3.5549 Å and refinement parameters of $R_p$ = 4.95% and $R_{wp}$ = 5.89%. In addition, the negligible difference plot of JNM-7-AA also suggested a good agreement between the experimental data and the refined PXRD patterns. JNM-8-AA featured an eclipsed configuration and crystallized in the P6/m space group (Fig. 2b and Supplementary Fig. 5). In sharp contrast, the PXRD patterns of JNM-9-ABC displayed six peaks at 2.88°, 5.80°, 8.82°, and 11.58° from the (110), (220), (101), and (330) diffractions in the calculated ABC stacking mode, respectively (Fig. 2c). Pawley refinements of JNM-9-ABC gave a trigonal space group R-3 with an optimized unit cell of a = b = 56.1215 Å and c = 10.2004 Å and refinement parameters of $R_p$ = 2.87% and $R_{wp}$ = 3.98% (Fig. 2c and Supplementary Fig. 10).

The chemical structure of JNMs was confirmed by Fourier transform infrared (FT-IR) and $^{13}$C CP/MAS NMR analysis. The FT-IR spectra of JNMs (Fig. 2d) revealed that the peak at 1670 cm$^{-1}$ assigned to the C = O stretching vibration in Cu-CTU and the N − H stretching vibrations located at 3210−3432 cm$^{-1}$ in di-amine linkers vanished. In addition, signals at around 1624−1628 cm$^{-1}$ attributed to the C = N stretching feature appeared. These results suggested the successful formation of imine bonds, and the Schiff-base condensation entirely proceeded. Furthermore, the $^{13}$C CP/MAS NMR spectra of JNMs (Fig. 2e) showed the disappearance of the aldehyde carbon signal at 184 ppm. In comparison, characteristic resonance peaks of the imine carbon at ~153 ppm appeared, further confirming the formation of the

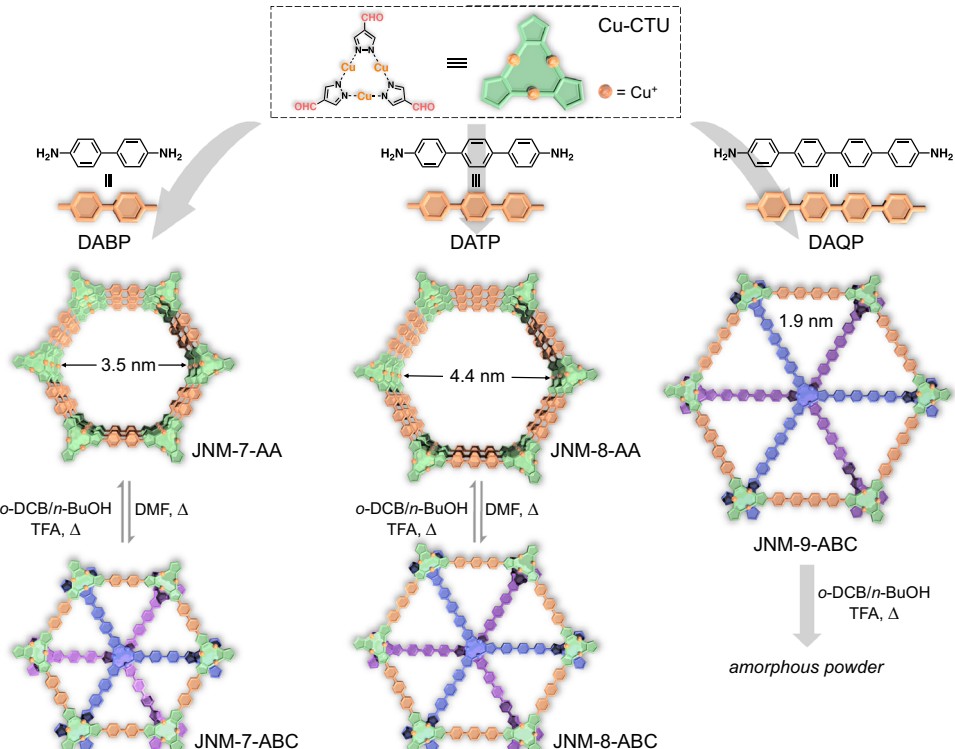

**Fig. 1 | Design of JNMs and the reversible structural transformation triggered by solvent, acid, and heat.** Top, Schematic illustration of the preparation of JNM-7-AA, JNM-8-AA and JNM-9-ABC. Bottom, Schematic representation of reversible interlayer stacking modulation between AA and ABC mode.

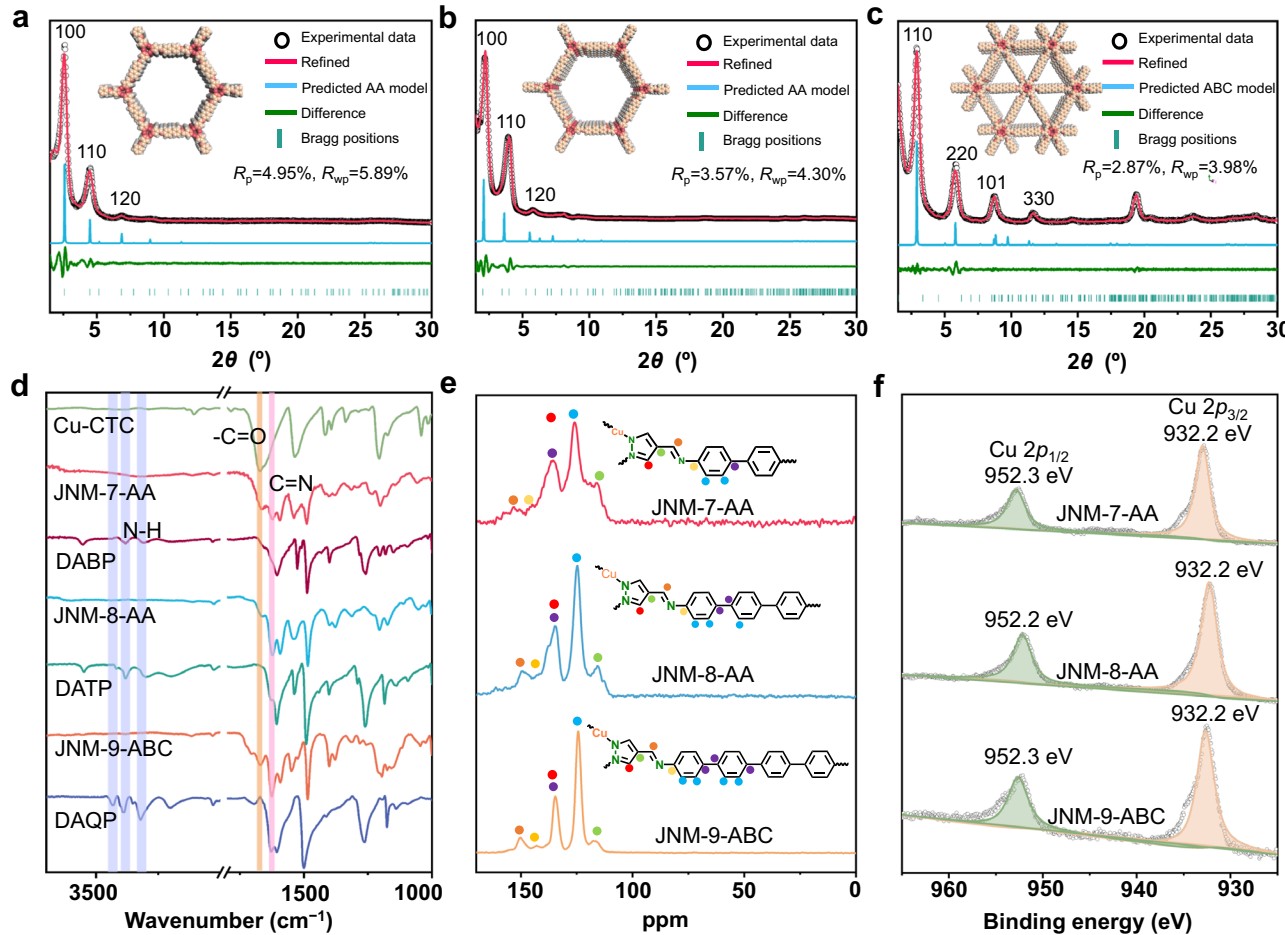

**Fig. 2 | Characterization of JNM-7-AA, JNM-8-AA and JNM-9-ABC.** Crystal structure of (**a**) JNM-7-AA, (**b**) JNM-8-AA, and (**c**) JNM-9-ABC showing space-filling model (C, light pink; H, white; Cu, red; N, pink). **d** FT-IR spectra of JNM-7-AA, JNM-8-AA, and JNM-9-ABC, Cu-CTU, and di-amine linkers. **e** Solid-state $^{13}$C CP-MAS NMR spectra (100 MHz, 300 K) of JNM-7-AA, JNM-8-AA and JNM-9-ABC. **f** XPS spectra of Cu 2$p$ for JNM-7-AA, JNM-8-AA and JNM-9-ABC.

imine bond. The X-ray photoelectron spectroscopy (XPS) spectra of JNMs showed that a symmetrical Cu(I) 2$p_{3/2}$ signal at 932.3 eV without satellite peaks was observed, indicating the oxidation state of copper ions in JNMs was monovalent (Fig. 2f). Scanning electron microscopy (SEM) images of JNM-7-AA, and JNM-8-AA both exhibited needle-like micro-crystals with micrometer size (Fig. 3a, c). In contrast, JNM-9-ABC showed nano-flakes morphologies composed of stick-like micro-crystals with nanometer size (Fig. 3e). The high-resolution transmission electron microscopy (HR-TEM) and fast Fourier transform of JNM-7-AA, JNM-8-AA, and JNM-9-ABC demonstrated that the well-ordered lattice fringe with the d-spacing of 3.10, 3.70, and 3.10 nm, corresponding to the lattice planes of (100), (100) and (110), respectively (Fig. 3b, d, f). This result is in good agreement with their refined PXRD pattern. Energy dispersive X-ray spectroscopy (EDS) of the JNMs revealed a uniform distribution of elements Cu, C, and N within the skeleton (Supplementary Figs. 11–13).

**Reversible interlayer structure transformation**
The reversible structure transformation of JNM-7-AA and JNM-8-AA can be triggered by the alteration of solvent and acid. Generally, pale-yellow crystalline powder of JNM-7-ABC or JNM-8-ABC could be obtained when a DMF solution of JNM-7-AA or JNM-8-AA was heated at 80 °C for 10 h (See SI for details). The PXRD patterns of these transformed samples were completely different from their parent sample (Fig. 4a, and Supplementary Fig. 19). Taking JNM-7-ABC as an example, it displayed the PXRD peaks at 4.20°, 8.28° and 12.38° for the (110), (220) and (330) reflection planes, respectively, which matched well

with ABC stacking model (Fig. 4a). Pawley refinement of JNM-7-ABC afforded a space group of R-3 with unit cell parameters of $a = b = 41.1837$ Å and $c = 10.2320$ Å and refinement parameters of $R_p = 3.03\%$ and $R_{wp} = 3.98\%$. Similarly, JNM-8-ABC showed four observed PXRD peaks at 3.36°, 6.68°, 10.16° and 13.58° assigned to (110), (220), (330), and (440), respectively. This was in good agreement with the simulated ABC stacking model with unit cell parameters of $a = b = 48.6710$ Å and $c = 10.2001$ Å and refinement parameters of $R_p = 3.32\%$ and $R_{wp} = 4.65\%$ (Supplementary Fig. 19). However, JNM-9-ABC remained intact when immersed in DMF (Supplementary Fig. 20). Interestingly, brown crystalline powders of JNM-7-AA or JNM-8-AA could be regenerated when JNM-7-ABC or JNM-8-ABC was added to a mixed solution of o-DCB, n-BuOH and TFA (v/v/v. 0.5/0.5/0.1) and heated at 80 °C for 10 h. The PXRD patterns of regenerated JNM-7-AA or JNM-8-AA were identical to the as-synthesized ones (Fig. 4b and Supplementary Fig. 21), confirming the reversible structure transformation process.

The FT-IR and $^{13}$C CP/MAS NMR spectra of JNM-7-ABC or JNM-8-ABC show similar characteristic peaks as JNM-7-AA or JNM-8-AA (Supplementary Figs. 24 and 25), confirming they have identical chemical structures. The SEM images of JNM-7-ABC and JNM-8-ABC show needle-like micro-crystals with much smaller sizes compared to JNM-7-AA and JNM-8-AA (Supplementary Fig. 26). The XPS spectra of JNM-7-ABC and JNM-8-ABC reveal that Cu$^+$ ions remain intact even after interlayer structure transformation (Supplementary Fig. 27). The thermal and chemical stability of the JNMs were investigated by thermogravimetric analysis (TGA) and variable-temperature PXRD. JNM-7-AA, JNM-8-AA, and JNM-9-ABC did not show noticeable weight loss,

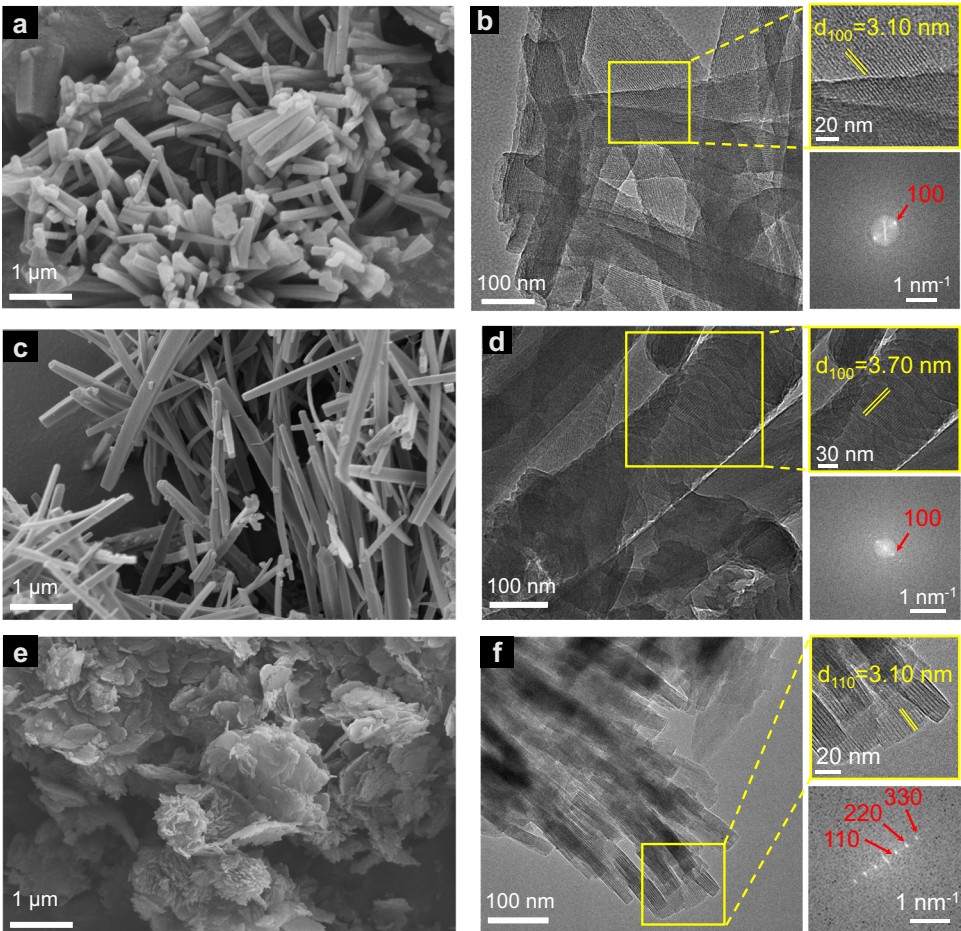

**Fig. 3 | SEM and TEM imagrs of JNM-7-AA, JNM-8-AA and JNM-9-ABC.** SEM images of (**a**) JNM-7-AA, (**c**) JNM-8-AA, and (**e**) JNM-9-ABC. HR-TEM images of (**b**) JNM-7-AA, (**d**) JNM-8-AA, and (**f**) JNM-9-ABC Top right: enlarged images of a selective area showing well-ordered lattice fringe. Bottom right: fast Fourier transform (FFT) pattern.

and their crystallinity remained up to ~300 °C, while JNM-7-ABC and JNM-8-ABC were only stable up to ~200 °C (Supplementary Figs. 28 and 29). All JNMs are stable in common organic solvents, including methanol, acetonitrile, dichloromethane, and chloroform, as well as acid (i.e., 0.1 M HCl) and base (i.e., 0.1 M NaOH) (Supplementary Fig. 30).

Since the interlayer sliding will primarily affect the surface area and pore size, the porosity of JNM-7 and JNM-8 was examined by nitrogen adsorption measurements at 77 K to further verify the reversible structure transformation. The freshly prepared JNM-7-AA and JNM-8-AA demonstrated type IV adsorption curves featuring a mesoporous nature, and the Brunauer−Emmett−Teller (BET) surface areas were calculated to be 1138.36 and 210.70 $m^2\,g^{-1}$ (Fig. 4c, and Supplementary Fig. 31), respectively. In addition, Nonlocal density functional theory (NLDFT) suggests a narrow pore-size distribution of JNM-7-AA or JNM-8-AA with an average pore width of about 3.4 or 4.0 nm (Fig. 4d and Supplementary Fig. 31), respectively, which were closed to the simulated values from the eclipsed AA mode (~3.5 and 4.4 nm for JNM-7-AA and JNM-8-AA, respectively). After the addition of DMF, the surface areas of JNM-7-ABC and JNM-8-ABC were remarkably decreased to 157.90 $m^2\,g^{-1}$ and 122.49 $m^2\,g^{-1}$ (Fig. 4c and Supplementary Fig. 31), respectively. Meanwhile, the pore-size distribution of JNM-7-ABC and JNM-8-ABC also significantly declined to ~1.4 and ~1.5 nm (Fig. 4d, and Supplementary Fig. 31), further supporting the structural transformation from AA to ABC stacking mode. After soaking of JNM-7-ABC and JNM-8-ABC in a TFA solution, the BET surface areas (744.42 and 178.64 $m^2\,g^{-1}$) and pore size distribution (3.0 and 3.9 nm) of

regenerated JNM-7-AA and JNM-8-AA were close to those of the pristine samples, strongly confirming the reversible structure transformation from the ABC to AA stacking mode (Fig. 4c, d and Supplementary Fig. 31).

To further elucidate the interlayer shifting processes, the density functional theory (DFT) calculations were conducted on the JNM-7 system using the DMol[3] molecular dynamics module (Fig. 4e, f). Six processes were considered in the calculation: (I) the interaction between DMF molecules and JNM-7-AA; (II) the interaction between DMF molecules and JNM-7-ABC; (III) the removal of DMF from JNM-7-ABC; (IV) the interaction between TFA molecules and JNM-7-ABC; (V) the interaction between TFA molecule and JNM-7-AA; (VI) the removal of TFA from JNM-7-AA. As shown in Fig. 4e, the energy of JNM-7-ABC in DMF was much higher than that of JNM-7-AA, suggesting ABC stacking was thermodynamics unfavored state in DMF and the energy input was required (i.e., heating) to trigger the structure transformation. These results are consistent with the experimental observations, in which JNM-7-AA in DMF cannot transfer to JNM-7-ABC without heating. In addition, the energy of JNM-7-AA in TFA is much smaller than that of JNM-7-ABC, suggesting AA stacking is thermodynamics preferred.

### Reversible encapsulation and release of lipase
The adsorption of enzymes by porous material is an important research field for developing advanced composite materials with advanced functions. However, achieving reversible adsorption and desorption of enzymes using COFs triggered by environmental stimulation is still hard[23–26]. Since lipase (from thermophilic bacteria) has

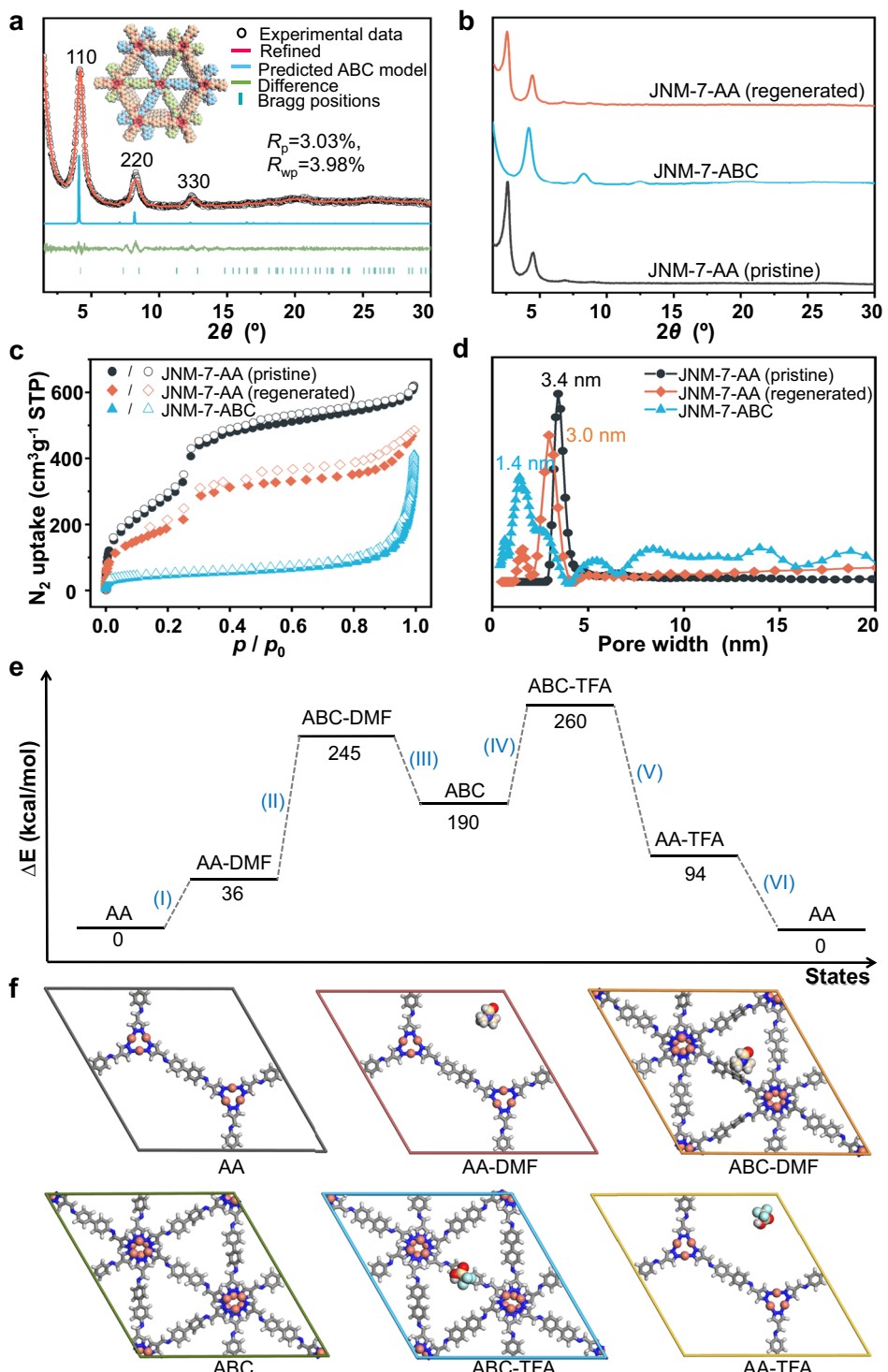

**Fig. 4 | Reversible structure transformation of JNM-7. a** crystal structure of JNM-7-ABC showing space-filling model (C, light pink; H, white; Cu, red; N, pink). **b** PXRD analysis, (**c**) BET surface analysis, and (**d**) pore size distribution profiles of JNM-7-AA (pristine), JNM-7-ABC, and JNM-7-AA (regenerated). **e** DFT-calculated energy landscape and (**f**) the optimized structures of JNM-7 under different states: (AA) initial AA stacking, (AA-DMF) interaction with DMF in AA stacking, (ABC-DMF) interaction with DMF in ABC stacking, (ABC) dried state in ABC stacking, (ABC-TFA) interaction with TFA in ABC stacking, (AA-TFA) interaction with TFA in AA stacking.

a size of 3.5 nm × 3.0 nm × 4.3 nm, which is smaller than the pore size of JNM-8-AA (4.0 nm) but more significant than that of JNM-8-ABC (1.5 nm), we attempt to demonstrate the reversible encapsulation and release of lipase via reversible structure transformation process of JNM-8. To a phosphate buffer solution (PBS) of lipase (30 mg mL$^{-1}$, pH = 7.0), JNM-8-AA (15 mg) was added, and the resulting mixture was shaken at 175 rpm. After centrifugation and filtration, Ultraviolet

−visible (UV−vis) spectroscopy of collected filtrate was recorded to monitor the change of concentration of lipase using the *n*-butyl cyanoacrylate (BCA) method (Fig. 5a)[27−29]. JNM-8-AA can absorb 24.80 mg lipase, denoted to LP@JNM-8-AA, within 6 h, showing an adsorption efficiency of 82.7% (Fig. 5b, c). In addition, the installed lipase can be released via the interlayer structure transformation. Specifically, after centrifugation, the collected powder of LP@JNM-8-AA was added to a

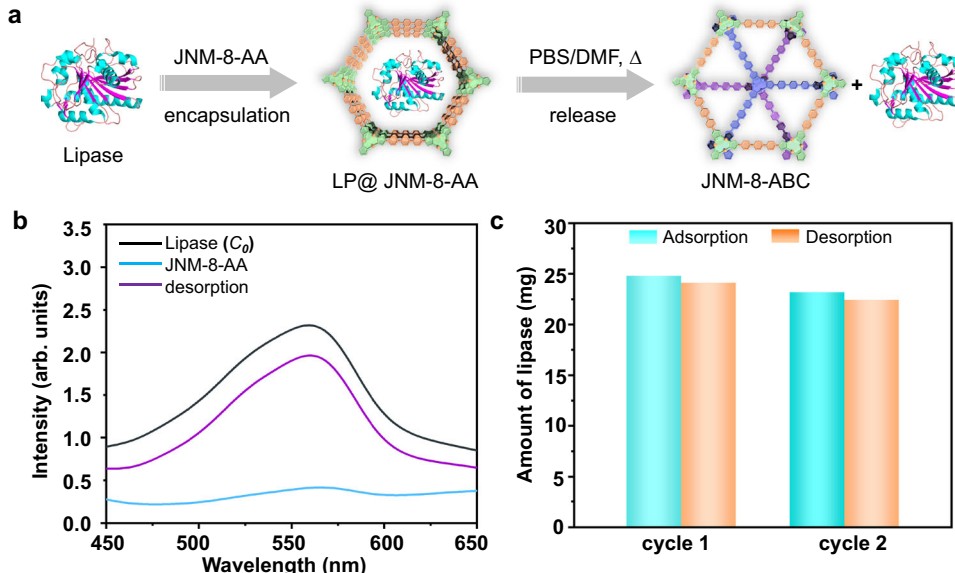

**Fig. 5 | Reversible encapsulation and release of lipase. a** Schematic illustration of reversible encapsulation and release of lipase. **b** UV−vis absorption spectra of the solution of lipase. black line: original solution containing 30 mg of lipase; light blue line: after treatment with JNM-8-AA for 6 h; purple line: LP@JNM-8-AA after treatment with DMF for 6 h. **c** Reversibility test of JNM-8-AA for adsorption (cyan) and desorption (orange) of lipase.

solution of PBS and DMF. The resulting mixture was heated at 80 °C for 6 h. After then, lipase (24.14 mg) was released, and the desorption efficiency was estimated to be 97.3%, as confirmed by the UV–vis analysis (Fig. 5b, c). The PXRD experiment of the resulting powder suggested that JNM-8-AA ultimately transferred to JNM-8-ABC (Supplementary Fig. 33). Such reversible adsorption and desorption processes can be repeated at least twice with a slight decrease in efficiency in the second cycle (i.e., adsorption and desorption efficiency of 77.3% and 96.7%, respectively) (Supplementary Fig. 34).

## Optical and electronic properties

Besides the tailoring porosity, the alteration of interlayer stacking of JNM-7 and JNM-8 greatly affected their optical and electronic properties. As shown in Fig. 6, the solid-state UV-Vis diffuse reflectance spectroscopy of JNM-7-AA exhibited a strong absorption band range from 200 nm to 450 nm. In contrast, the absorption edge of JNM-7-ABC was located at 380 nm, indicating the AA stacking model delivered broader light absorption range than that of ABC stacking mode. Consequently, the optical bandgap ($E_g$) of JNM-7-AA was estimated to be 2.18 eV by the Tauc plot, which was much narrower than that of JNM-7-ABC (2.63 eV) (Fig. 6a). Similar to JNM-7, JNM-8-AA possessed a wider absorption range (200−450 nm) and a narrower $E_g$ of 2.18 eV than those of JNM-8-ABC (200−400 nm and 2.55 eV) (Fig. 6b). In addition, the flat band potentials of JNM-7-AA, JNM-7-ABC, JNM-8-AA, and JNM-8-ABC were determined to be −1.178, −0.735, −0.857, and −1.620 eV vs. Ag/AgCl, respectively (Supplementary Fig. 35), which were equal to their conduction band (CB) potentials, by the Mott−Schottky experiments. Afterward, their valence band (VB) potentials could be calculated to be 1.202, 2.095, 1.523, and 1.501 eV vs. NHE, respectively (Fig. 6c), by a combination of the Mott−Schottky experiments and its optical bandgap data. These results suggest that the AA stacking model has a broader light absorption and narrower $E_g$ than the corresponding ABC stacking model. This could be attributed to an eclipsed AA stacking model has the stronger π-π and Cu-Cu interactions between layers compared to the corresponding ABC stacking model. Moreover, to further evaluate the photo-electrochemical properties of JNM-7 and JNM-8, the transient photocurrent measurements and the electrochemical impedance spectroscopy (EIS) were conducted. Their transient photocurrent intensity follows the order of

JNM-8-AA > JNM-7-AA > JNM-8-ABC > JNM-7-ABC. Importantly, the photocurrent intensity of JNM-8-AA and JNM-7-AA are ~1 time larger than those of JNM-8-ABC and JNM-7-ABC (Fig. 6d), respectively, suggesting the spatial separation of photogenerated charge carriers in AA stacking model is more effective compared to the corresponding ABC stacking model. Furthermore, JNM-8-AA and JNM-7-AA delivered similar charge transfer resistances, which were much lower than those of JNM-8-ABC and JNM-7-ABC (Fig. 6e), indicating the higher charge-separation efficiency in the AA stacking model compared to ABC isomer.

The thermodynamically suitable $E_g$ of JNM-7 and JNM-8 for photocatalytic reduction of $O_2$ to $O_2^{\cdot-}$ (Fig. 6c)[30–33] encouraged us to investigate their photogenerated $^1O_2$ and $O_2^{\cdot-}$ capability. To accomplish this, electron paramagnetic resonance (EPR) measurements were conducted. By addition of a radical trapping reagent, 5,5-dimethyl-1-pyrroline N-oxide (DMPO), the intense signals of $O_2^{\cdot-}$ appeared upon the light irradiation of JNM-7-AA and JNM-8-AA in air (Fig. 6f). In addition, as shown in Supplementary Fig. 37, in the presence of 2,2,6,6-tetramethyl-4-piperidone (TEMP), a $^1O_2$ trapping reagent, the characteristic signals of TEMPO were observed under photo-irradiation of JNM-7-AA and JNM-8-AA in air. In sharp contrast, only weak signals of $O_2^{\cdot-}$ or TEMPO were observed using JNM-7-ABC and JNM-8-ABC as photosensitizers under the same conditions (Fig. 6f). These results illustrated that JNM-7 and JNM-8 in AA stacking model have much higher photocatalytic activity for sensitizing $O_2$ to $^1O_2$ and $O_2^{\cdot-}$ than their ABC isomers.

## Heterogeneous photocatalysis

Due to their favorable optical and electronic properties, JNM-7 and JNM-8 are potential photocatalysts for oxidative cross-coupling reactions. We initially tried the photo-induced cross-dehydrocoupling (CDC) reaction of N-phenyl-tetrahydroisoquinoline (**1**) and phenylacetylene (**2a**), and C-1 substituted tetrahydroisoquinoline derivatives, which were important drug motifs and exhibited a variety of biological activities[34–39], can be synthesized. To our delight, the coupling product (**3a**) can be obtained in 97% isolated yield using JNM-8-AA as a photocatalyst (Table 1, entry 1). Subsequently, we optimized the reaction conditions (Table 1, entries 2−16). Specifically, the mixture of **1** (0.1 mmol), **2a** (0.1 mmol), and JNM-8-AA (2.5 mol%) as a catalyst in

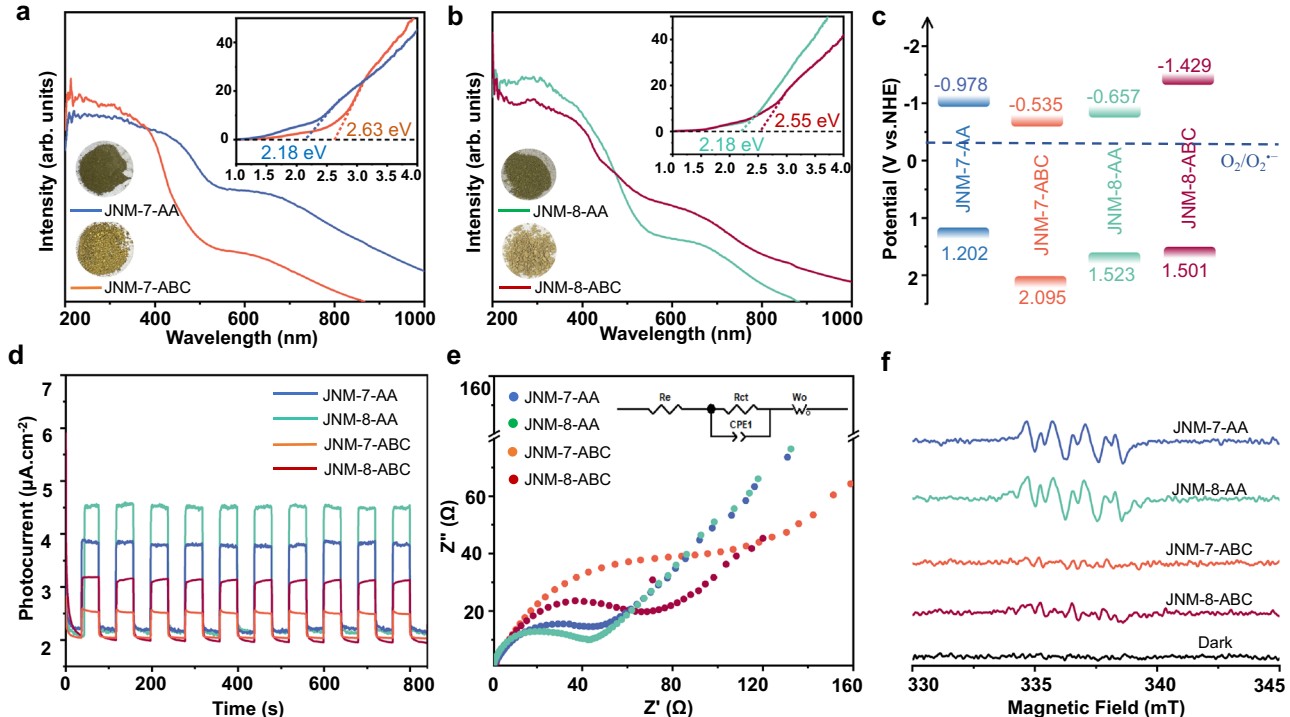

**Fig. 6 | Optical and electronic properties of JNMs.** The solid-state UV-Vis diffuse reflectance spectroscopy of (**a**) JNM-7-AA and JNM-7-ABC. **b** JNM-8-AA and JNM-8-ABC. Inset, the photographs of powder samples of JNMs. **c** Energy level diagrams of JNMs. **d** Photocurrent response curves of JNMs. **e** Nyquist plots of EIS of JNMs. Inset, equivalent circuit diagram. **f** the EPR spectra of JNMs (5 mg) in 3 mL of $CH_3CN$ with DMPO under air atmosphere in the dark or upon white LED light irradiation for 5 min.

$CH_3CN$ at room temperature (rt) under photo-irradiation with 12 W white LED for 12 h, affording **3a** in a > 99% conversion confirmed by GC-MS spectra (Table 1, entry 1). JNM-7-AA gave a slightly lower yield of 87%, while JNM-7-ABC and JNM-8-ABC delivered remarkably lower yields of <1% and 20%, respectively, due to their much lower capacity for sensitizing $O_2$ to $^1O_2$ and $O_2^{\cdot-}$ (Table 1, entries 2−4).

The reaction cannot proceed in the absence of JNMs, light irradiation, and $O_2$ (Table 1, entries 5-7). Changing the solvent to $CHCl_3$, DMF, and toluene also remarkably reduced the yields to 45%, 1%, and 7%, respectively (Table 1, entries 8-10). Reduction of JNM-8-AA loading to 1 mol% will also reduce the yield to 60% (Table 1, entry 11). No product was observed when linker DABP or DATP was employed as a catalyst instead of JNM-8-AA (Table 1, entries 12−13). Various copper-based catalysts were employed, a yield of 85%, 19%, and 13% were obtained, implying the Cu-CTU is crucial for the photocatalytic CDC reactions (Table 1, entries 14−16). A large-scale reaction with 1 g of **1** and low loading of JNM-8-AA was further performed, and then 0.92 g of **3a** was obtained with a yield of 60%. The turnover frequency (TOF) is estimated to be ~36 h$^{-1}$, which is faster compared to reported representative catalysts (Supplementary Table 13). To further study the reusability of catalyst, JNM-8-AA was collected after the completion of the CDC reaction and reused for the next catalytic cycle with the addition of a fresh reaction solution. Interestingly, catalytic performance of JNM-8-AA did not show noticeable decrease after three catalytic cycles (Supplementary Fig. 40). More importantly, the crystallinity of JNM-8-AA remained and Cu(I) ions in JNM-8-AA were unchanged after three catalytic cycles confirmed by PXRD and XPS analysis (Supplementary Figs. 40 and 41).

To understand the reaction mechanism, control experiments were conducted. CDC product of **3a** did not observed after adding an $O_2^{\cdot-}$ quench reagent (i.e., 1,4-benzoquinone (BQ)). However, a 90% yield was obtained upon the introduction of TEMPO, a quencher of $^1O_2$ (Supplementary Fig. 42). These results suggested the photo-induced CDC reaction is driven by $O_2^{\cdot-}$ rather than $^1O_2$, which is similar to reported examples[40,41]. Thus, a reaction mechanism is proposed as shown in Supplementary Fig. 43. We further screened the substrate scope of substituted terminal alkyne under standard catalytic conditions (Table 2). As shown in Table 2, electron-donating and electron-withdraw substituents (**2a-2d**), aliphatic alkynes (**2e-2g**), and tri-methylsilyl groups (**2 h**) can be tolerant and give good yields of CDC products from 80% to 97%, suggesting JNM-8-AA a promising photocatalyst for CDC reactions.

## Discussion

In summary, we have successfully synthesized three Cu-CTU-based CuOFs, denoted to JNM-n (n = 7, 8, and 9), through imine condensation between Cu-CTU and three diamine linkers. Upon environmental stimulation, JNM-7-AA and JNM-8-AA can reversibly transfer to JNM-7-ABC and JNM-8-ABC, respectively, enabling the reversible adsorption and desorption of lipase. Due to the control of interlayer interactions and stacking mode, JNM-7-AA and JNM-8-AA exhibit a broader light absorption range, higher charge-separation efficiency, and better photocatalytic activity for sensitizing $O_2$ to $^1O_2$ and $O_2^{\cdot-}$ than their ABC stacking isostructure. In addition, JNM-8-AA exhibits much higher photocatalytic activities for oxidative cross-coupling reactions than JNM-8-ABC. Moreover, JNM-8-AA delivers good reusability and catalytic activity with a TOF of about 36 h$^{-1}$ for CDC reaction, much faster than many other reported homogeneous and heterogeneous catalysts.

## Methods
### General procedure
PXRD data was collected on a Rigaku Ultima IV diffractometer (40 kV, 30 mA). The parameters are as follows: scan speed (0.5° per min), step size (0.02°), and scan range (1.5°−30°). Thermogravimetric analysis was performed on a Mettler-Toledo (TGA/DSC1) thermal analyzer. The

**Table 1 | Screen reaction conditions for photoinduced CDC reaction**

| Entry | Change from the "standard condition" | Yield (%)[a] |
|---|---|---|
| 1 | none | 99 (97)[b] |
| 2 | JNM-7-AA | 87 |
| 3 | JNM-7-ABC | <1 |
| 4 | JNM-8-ABC | 20 |
| 5 | no JNMs | 0 |
| 6 | no $h\nu$ | 0 |
| 7 | under $N_2$, instead of air | <1 |
| 8 | $CHCl_3$, instead of $CH_3CN$ | 45 |
| 9 | DMF, instead of $CH_3CN$ | 1 |
| 10 | toluene, instead of $CH_3CN$ | 7 |
| 11 | 1 mol% JNM-8-AA | 60 |
| 12 | linker DABP, instead of JNM-8-AA | 0 |
| 13 | linker DATP, instead of JNM-8-AA | 0 |
| 14 | Cu-CTU, instead of JNM-8-AA | 85 |
| 15 | CuO, instead of JNM-8-AA | 19 |
| 16 | CuI, instead of JNM-8-AA | 13 |
| 17 | addition of 1,4-benzoquinone[c] | <1 |
| 18 | addition of TEMPO | 90 |
| 19[c] | 1 g scale | 60 |

Reaction conditions: **1a** (0.1 mmol), **2a** (0.1 mmol), JNM-8-AA (2.5 mol%), $CH_3CN$ (1 mL), room temperature (rt), white LED (12 W), air.
[a]Determined by GC-MS analysis.
[b]Isolated yields.
[c]JNM-8-AA ($6.9 \times 10^{-3}$ mmol) was used instead of 2.5 mol%.

SEM images and EDS were acquired on a JEOL JSM7600F microscope with an acceleration voltage of 5 kV. TEM analysis was conducted on an FEI Titan 80−300 S/TEM operated at 200 kV. A Nicolet Avatar 360 FT-IR spectrophotometer was used for conducting FT-IR spectroscopy. XPS were measured utilizing a Thermo ESCALAB 250XI system. GC-MS analysis was carried out on an Agilent 7890B GC analyzer. The $^1H$ and $^{13}C$ NMR spectra were recorded on Bruker Biospin Avance (400 MHz) equipment. Solid-state NMR experiments were conducted on a Bruker WB Avance II 400 MHz NMR spectrometer. Gas sorption analyses were preformed on an ASAP 2020 PLUS Analyzer (Micromeritics). Surface areas and pore size distribution profiles were determined using Brunauer-Emmett-Teller methods and the DFT method, respectively. All chemicals were used without further purification and purchased from commercial sources.

## Synthesis of JNMs
**General procedure.** Di-amine linkers (i.e., DABP, DATP or DAQP, 0.075 mmol), Cu-CTU (23.7 mg, 0.05 mmol)[20], a mixed solvent of 1,2-dichlorobenzene and 1-butanol (1 mL, v- v = 7: 3, 1: 1 or 3: 7), and TFA (0.1 mL, 6 M) were charged to a Schlenk tube (10 mL). Afterward, the mixture was frozen and degassed with three freeze-pump-thaw cycles using liquid nitrogen. Upon warming to rt, the mixture was heated at 120 °C for 3 days. The crystalline powder was collected by filtration, and then, the crystalline solid was washed with methanol, ethanol and

dichloromethane. The resulting solid was further dried at 100 °C under vacuum for 8 h to give JNMs.

**JNM-7-AA.** According to general procedure, *p*-4,4'-diaminobiphenyl (DABP) (13.8 mg, 0.075 mmol), 1 mL mixed solution of 1,2-dichlorobenzene and 1-butanol (v: v = 7: 3) was used and JNM-7-AA was obtained as brown powders. Yield: 21.5 mg (70.6% based on Cu). IR (KBr): v = 1668 (m), 1625 (s), 1592 (m), 1541 (m), 1488 (s), 1403 (w), 1376 (w), 1315 (w), 1201 (m), 1139 (w), 1062 $cm^{-1}$ (w). Elemental analysis for $C_{60}H_{42}N_{18}Cu_6 \cdot 2C_6H_4Cl_2 \cdot 11H_2O$, calcd (%): C 45.79, H 3.84, N 13.35; Found (%): C 42.04, H 2.80, N 13.82.

**JNM-8-AA.** According to general procedure, *p*-4,4'-diamino-*p*-terphenyl (DATP) (19.5 mg, 0.075 mmol), 1 mL mixed solution of 1,2-dichlorobenzene and 1-butanol (v: v = 1: 1) was used and JNM-8-AA was obtained as brown powders. Yield: 35.0 mg (97% based on Cu). IR (KBr): v = 1666 (m), 1625 (s), 1589 (m), 1545 (m), 1483 (s), 1393 (w), 1375 (w), 1316 (w), 1207 (m), 1171 (m), 1058 (m), 1002 $cm^{-1}$ (m). Elemental analysis for $C_{78}H_{54}N_{18}Cu_6 \cdot 3C_6H_4Cl_2 \cdot 10H_2O$, calcd (%): C 51.34, H 3.86, N 11.23; Found (%): C 48.67, H 2.12, N 11.13.

**JNM-9-ABC.** According to general procedure, 4,4'-diamino-*p*-quaterphenyl (DAQP) (25.2 mg, 0.075 mmol), 1 mL mixed solution of 1,2-dichlorobenzene and 1-butanol (v: v = 3: 7) was used and JNM-9-ABC

**Table 2 | Substrate Scope for CDC reaction photocatalyzed with JNM-8-AA.[a]**

3a: 97%    3b: 97%    3c: 90%    3d: 91%

3e: 80%    3f: 97%    3g: 85%    3h: 95%

[a]Isolated yields

was obtained as pale-yellow powders. Yield: 38.5 mg (92% based on Cu). IR (KBr): ν = 1670 (m), 1623 (s), 1589 (m), 1551 (m), 1487 (s), 1404 (w), 1323 (w), 1256 (w), 1193 (m), 1045 cm$^{-1}$ (w). Elemental analysis for $C_{96}H_{66}N_{18}Cu_6 \cdot C_6H_4Cl_2 \cdot 2H_2O$, calcd (%): C 60.17, H 3.66, N 12.38; Found (%): C 60.37, H 3.39, N 11.81.

**Reversible structure transformation**

**Structure transformation from AA to ABC.** To a 10 mL Schlenk tube, a pristine sample of JNM-7-AA (50 mg) or JNM-8-AA (50 mg) and 5 mL DMF was added. The resulting mixture was stirred at 80 °C for 10 h, and then the solid was isolated by filtration, washed, and solvent exchanged with EtOH and acetone several times. The resulting powder was dried under vacuum at room temperature for 3 h. Finally, the pale-yellow powders JNM-7-ABC or JNM-8-ABC were obtained. JNM-7-ABC IR (KBr): ν = 1669 (m), 1624 (s), 1594 (m), 1543 (m), 1489 (s), 1406 (w), 1376 (w), 1306 (w), 1241 (m), 1202 (s), 1241 (m), 1056 cm$^{-1}$ (m). JNM-8-ABC IR (KBr): ν = 1669 (m), 1625 (s), 1589 (m), 1532 (m), 1484 (s), 1404 (w), 1376 (w), 1320 (w), 1201 (s), 1175 (w), 1056 cm$^{-1}$ (m).

**Structure transformation from ABC to AA.** JNM-7-ABC (20 mg) or JNM-8-ABC (20 mg), 0.5 mL of 1,2-dichlorobenzene, 0.5 mL of 1-butanol, and 0.1 mL of TFA (6 M) were added a 10 mL Schlenk tube. The resulting mixture was stirred at 80 °C for 10 h, and then the solid was isolated by filtration, washed, and solvent exchanged with EtOH and acetone several times. The resulting powder was dried under vacuum at room temperature for 3 h. Finally, the brown powder JNM-7-AA or

JNM-8-AA with similar diffraction peaks to the AA simulation was obtained.

**Reversible encapsulation and release of lipase**

**Encapsulation of lipase.** In a typical adsorption experiment, 15 mg of JNM-8-AA was added to 1 mL of an aqueous phosphate buffer solution of lipase (pH = 7.0, $C_0$ = 30 mg/mL), and the mixture was shaken at 175 rpm and 25 °C for 6 h. Afterward, the resultants were centrifuged, and the solid was collected to give LP@JNM-8-AA. The supernatant was further analyzed by UV-Vis spectroscopy using the n-butyl cyanoacrylate (BCA) method to provide the concentration of lipase ($C_{at}$)[27,28].

The adsorption efficiency of lipase was calculated as following equation (1):

Adsorption efficiency (%) = ($C_0$ − $C_{at}$) / $C_0$ × 100 (1)

$C_0$ and $C_{at}$ are the lipase concentrations at the initial condition and in the filtrate after adsorption, respectively.

**Release of lipase.** The LP@JNM-8-AA obtained in the above adsorption process was added to an aqueous phosphate buffer (0.5 mL), and DMF mixture solution (0.5 mL), and then the mixture was heated at 80 °C for 6 h, followed by cooling down to room temperature. Afterward, the resultants were centrifuged and filtrated, and the solid was collected to give JNM-8-ABC. The filtrate was further analyzed by UV-Vis spectroscopy using the n-butyl cyanoacrylate (BCA) method to provide the concentration of lipase ($C_{dt}$).

The desorption efficiency of lipase was calculated as following equation (2):

Desorption efficiency (%) = $C_{dt}$ / ($C_0 - C_{at}$) × 100 (2)

$C_0$, $C_{at,}$ and $C_{dt}$ are the lipase concentration at the initial condition, in the filtrate after the adsorption process, and in the filtrate after the desorption process, respectively.

**Recyclability test.** After one run of adsorption and desorption, JNM-8-ABC obtained from the above desorption process was treated with TFA. The solid was isolated by filtration and washed with ultra-pure water and methanol. The resultant solid was dried under vacuum at 80 °C for 6 h to give regenerated JNM-8-AA, which was used for another adsorption and desorption experiment.

**DFT calculations.** All calculations were performed by using the density functional method conducted out by the DMol[3] molecular dynamics module embedded in Materials Studio 2018 (MS 2018). Generalized gradient approximation (GGA) and Perdew–Burke–Ernzerhof (PBE) were used. The total energy difference and maximum residual force converged within $10^{-4}$ Ha and 0.05 Ha/Å during optimization. In all calculations, we used periodic boundary conditions and a supercell large enough to present the full coordination environments of JNM-7. The simulated AA-stacked and ABC-stacked JNM-7 structures were optimized first; solvent molecules were introduced to the channel pore.

**General procedure for the CDC reaction.** The JNM-8-AA were activated in a vacuum at 120 °C for 8 h prior to use for the catalytic experiment. A mixture of **1** (19.03 μL, 0.1 mmol), alkynes (**2a-h**) (0.1 mmol), JNM-8-AA (1.82 mg, 0.0025 mmol) in $CH_3CN$ (1 mL) was stirred at rt, and the resulting mixture was irradiated for 12 h utilizing 12 W white LED. Afterward, the supernatant was analyzed by GC–MS, and the reaction conversion was estimated based on the alkynes.

### Reporting summary

Further information on research design is available in the Nature Portfolio Reporting Summary linked to this article.

## Data availability

The data that support the findings of this study are available within the paper and its supplementary information files or are available from the corresponding authors upon request.

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

## Acknowledgements

G.H.N. is thankful for the financial support from Guangdong Basic and Applied Basic Research Foundation (no. 2019B151502024) and Guangzhou Science and Technology Project (202201020038). This study was supported financially by the National Natural Science Foundation of China (nos. 22371091, 21975104, 22150004, and 21731002), the Guangdong Major Project of Basic and Applied Research (no. 2019B030302009). Q.G. is thankful for the financial support from Special Project for Peak Carbon Dioxide Emissions-Carbon Neutrality (21DZ1206900) from the Shanghai Municipal Science and Technology Commission. The authors acknowledge Prof. Daqiang Yuan for the density functional theory (DFT) calculations. The authors acknowledge Dr. Heng Zeng for the dissucssion of BET analysis and Dr. Zhongxin Chen for the English editing of the manuscript.

## Author contributions

G.H.N and D.L. designed the research; P.Y.Y. and K.M.M. conducted the experiments and data analysis; Y.M.W., X.C.L., K.M.M., J.T.L. and R.J.W. contributed to material analysis; M.X. contributed to DFT calculation; Q.G. contributed to the BET analysis; P.Y.Y., G.H.N. and D.L. co-wrote the manuscript. All authors read and commented on the manuscript.

## Competing interests

The authors declare no competing interests.
