## [Peer review file · Nature Communications]

REVIEWER COMMENTS

Reviewer #1 (Remarks to the Author):

The authors report a modulation method to manipulate the interlayer stacking of two copper-containing covalent organic frameworks through solvent and heating stimulation. While this topic is of interest, it has been reported many times by prior publications. For example, Dan Zhao group has found that solvent induction and CO₂ adsorption or desorption could effectively tuning the layer stacking of 2D COFs (<https://pubs.acs.org/doi/10.1021/jacs.0c03691>; <https://pubs.acs.org/doi/abs/10.1021/jacs.2c08214>). In addition, many reports have already studied the ABC stacking and tuning previously (<https://onlinelibrary.wiley.com/doi/abs/10.1002/sml.202303684>; <https://pubs.acs.org/doi/pdf/10.1021/acsami.1c03170>; <https://pubs.acs.org/doi/10.1021/jacs.8b08452>; <https://onlinelibrary.wiley.com/doi/full/10.1002/anie.202114059>; <https://www.journal.csj.jp/doi/10.1246/cl.200834>; <https://www.sciencedirect.com/science/article/abs/pii/S0143720822008002>; <https://pubs.acs.org/doi/10.1021/jacs.8b07450>). And the photocatalytic application of COFs have been extensively studied in the area, without too much novelty. Therefore, the novelty of this manuscript is significantly decreased. The only difference in this manuscript is that the authors studied this problem based on two copper-containing COFs. Finally, there are several confusing parts in this manuscript which significantly decrease the soundness of the conclusion and need to be addressed. Therefore, I do not recommend it for publication in Nature Communications. It can be transferred to other journals after addressing the following concerns.

1. Figure 1: the bond between Cu and N should be coordination bond, so please use dash line instead of solid line.
2. Can the authors explain why JNM-9-ABC has a ABC stacking directly instead of having AA stacking?
3. Figure 2; the colors in Figure 2 are very difficult to distinguish. I advise to change the colors of these plots to make them clearer.
4. Figure 3: HRTEM measurements are much more important than SEM and low-resolution TEM, but in Figure 3, the authors only present the TEM images with crystal lattice in insets, which is very confusing. Please provide HR TEM data and present the electron diffraction mode.
5. Figure 4: how do the authors rule out the possibility that the topology of COFs changed rather than the change of layer stacking?
6. The N₂ sorption capacity of these materials are extremely low for all COFs, which might be resulted from the low crystallinity of these materials. Please improve the quality of these COFs.
7. Since the N₂ sorption is so low, is it accurate to determine the pore size distribution via such low sorption isotherms?
8. In Figure 4d, the difference of 2.26 and 1.2 is 1, and the difference of 2.95 and 2.26 is 0.7. Is it hard to say which is closer based on so minor difference. This is the same for JNM-8. Another question is that why does the pore size not go back to the original size? Is this because of partial transformation? If so, why are there no two peaks for the recovered samples? Please explain this.
9. The DFT calculation is very unreasonable. The DMF molecule in AA-stacking model is so close to the skeleton, while the DMF molecule in ABC-stacking model is adjacent to the skeleton. It is very apparent that closer distance will definitely result in higher energy levels. Additionally, the calculation does not make too much sense as these results are based on theoretical models. All DFT researchers know that ABC stacking will lead to higher energy levels than AA stacking regardless the materials, so the conclusion is very obvious and not surprising.
10. SEM EDS is not as accurate as TEM. Since the authors can get TEM measurements, why not use TEM EDS for elemental mapping?
11. Please provide elemental analysis data for these COFs in this work.
12. What is the possible mechanism for the claimed layer stacking? Is there any proof for this mechanism? Please clarify.

13. I noticed that this manuscript has many typos or grammar issues. Please check through very carefully and improve the readability of this work.

Reviewer #2 (Remarks to the Author):

The authors have prepared three 2D COFs composed from Cu-CTU and three di-amine linkers with different length. Interestingly, the interlayer stacking structure of JNM-7 and JNM-8 can be reversibly modulated, leading to the reversible encapsulation and release of lipase. Due to the alteration of stacking, AA stacking structure exhibited better photocatalytic activity for oxidative cross-coupling reaction. Obviously, it is very hard to achieve reversible structure transformation and obtain permanent AA and ABC structure with the identical components. Thus, these results are of great importance and deserve publication in such a reputed journal as Nat. Comm. Moreover, the work is solid, the materials are thoroughly characterized, and structural transformation mechanism is well investigated. I think it is a very good addition to COF and MOF area and am happy to see it published after addressing some points:

1. The ABC stacking with R3 space group should be trigonal, not hexagonal.
2. For the reversible interlayer structure transformation, did author try other solvent instead of DMF?
3. From the BET analysis, it seems the as synthesized JNM-7-AA have a smaller pore at 1.20 nm, did this can be attributed to ABC stacking model?
4. The For elemental analysis of JNMs, the calculated and observed ratio are not in good agreement with each other. It's better to re-check.

Reviewer #3 (Remarks to the Author):

In this paper, the authors reported the synthesis of a series of CuOFs, and their reversible structural transformation triggered by acid or heat. Due to the modulation of interlayer stacking model, the porosity and photocatalytic performance can be tuned. The materials and properties were convincingly and systematically studied including crystal structure, porosity, stability, host-guest chemistry and catalytic performance. In addition, the authors extensively investigated the reversible structural transformation and its mechanism. This paper is interesting and would be a valuable contribution to the field. Therefore, this reviewer recommends its publication in Nat. Comm., after minor revisions after the following comments are addressed.

1. The space group "P6/M" should be changed to "P6/m".
2. Graphics and texts are inconsistent. In Figure 5a, it shows that LP@JNM-8-AA was heated in DMF to release the LP. But in manuscripts, "....after centrifugation, the collected powder of LP@ JNM-8-AA was added to phosphate buffer solution and the resulting mixture was heated at 80°C for 6 h. Lipase(Figs. 5b, and 5c).". Please check and revise.
3. In the Supplementary Table 12, did the author calculate the TOF based on one Cu ion or one Cu-CTU (three Cu ions)?
4. In the Supplementary Figure 35, it should be the signal of 1O2 instead of O2•-.
5. The theoretical and experimental values of the elemental analysis of JNM-7-AA, JNM-8-AA and JNM-9-ABC are quite different.

Response Letter

Comments from Reviewers and Our Responses

(Black lines: original comment from the reviewers; Blue lines: author's response; Red line: author's action)

Reviewer 1 (Remarks to the Author):

The authors report a modulation method to manipulate the interlayer stacking of two copper-containing covalent organic frameworks through solvent and heating stimulation. While this topic is of interest, it has been reported many times by prior publications. For example, Dan Zhao group has found that solvent induction and CO₂ adsorption or desorption could effectively tuning the layer stacking of 2D COFs (<https://pubs.acs.org/doi/10.1021/jacs.0c03691>; <https://pubs.acs.org/doi/abs/10.1021/jacs.2c08214>). In addition, many reports have already studied the ABC stacking and tuning previously (<https://onlinelibrary.wiley.com/doi/abs/10.1002/sml.202303684>; <https://pubs.acs.org/doi/pdf/10.1021/acsami.1c03170>; <https://pubs.acs.org/doi/10.1021/jacs.8b08452>; <https://onlinelibrary.wiley.com/doi/full/10.1002/anie.202114059>; <https://www.journal.csj.jp/doi/10.1246/cl.200834>; <https://www.sciencedirect.com/science/article/abs/pii/S0143720822008002>; <https://pubs.acs.org/doi/10.1021/jacs.8b07450>). And the photocatalytic application of COFs have been extensively studied in the area, without too much novelty. Therefore, the novelty of this manuscript is significantly decreased. The only difference in this manuscript is that the authors studied this problem based on two copper-containing COFs. Finally, there are several confusing parts in this manuscript which significantly decrease the soundness of the conclusion and need to be addressed. Therefore, I do not recommend it for publication in Nature Communications. It can be transferred to other journals after addressing the following concerns.

Author's response: We thank this reviewer for reviewing our manuscript and for raising important questions to improve our work.

Zhao's group reported guest-triggered (e.g., solvent or CO₂) reversible interlayer shifting in 2D COFs prepared from the identical monomers (ref *J. Am. Chem. Soc.* 2020, 142, 12995–13002; *J. Am. Chem. Soc.* 2022, 144, 20363–20371), but the resulting *quasi-AA or -AB stacking structure cannot preserve in the absence of guests* (See Fig. R1a), leading to difficulty in comparing their properties. In addition, the interlayer stacking modes of 2D COFs can be adjusted *irreversibly* via ligand engineering or variation of reaction condition. (ref. *J. Am. Chem. Soc.* 2018, 140, 12922–12929; *J. Am. Chem. Soc.* 2018, 140, 16124–16133; *ACS Appl. Mater. Interfaces* 2021, 13, 29471–29481; *Small* 2023, 2303684). Therefore, compare to their work, we can obtain the stable AA and ABC structure with identical

components in a reversible manner (See Fig. R1b), allowing us to further reveal the interlayer stacking induced properties change such as adsorption and photocatalysis.

Overall, we would like to emphasize the importance and novelty of our work as follows:

1. Although Zhao's group has reported the guest-triggered reversible interlayer shifting, the resulting *quasi-AA or -AB stacking structure* are not stable when the guests were removed. In addition, no ABC stacking structure can be achieved. **JNM-7** and **JNM-8** are the first reported examples that can exhibited reversible interlayer shifting and the AA or ABC stacking isomer can preserve in the absence of guests. To further clarify this point, a schematic demonstration is shown in the Fig. R1.

Fig. R1. The schematic demonstration for tuning the interlayer stacking a) previous works, and b) this work.

2. Owing to reversible interlayer shifting, the reversible encapsulation and release of enzyme can be achieved, which have never been achieved before in the field of COFs. We believed that these unique properties would be applied in the drug delivery systems.

3. Although photocatalytic application of COFs has been extensively studied in the area, in our work, we intent to systematically study the relationship between stacking structure (i.e., AA and ABC isomers) and photocatalytic activities. Importantly, we have been found that AA stacking structure features broader light absorption range, higher charge-separation efficiency and higher photocatalytic activity

for CDC reaction than their ABC stacking isostructure. These observations are different from the reported COFs (*ACS Appl. Mater. Interfaces* 2021, 13, 29471–29481), indicating the introduction of Cu-CTU might be bring new functions into COFs. Such understanding could be significantly important for designing highly efficient COF-based photocatalysts. Moreover, **JNM-7-AA** delivered a high TOF for CDC reaction, which is much faster than many other reported homogeneous and heterogeneous catalysts.

1. Figure 1: the bond between Cu and N should be coordination bond, so please use dash line instead of solid line.

Author's action: Revised.

2. Can the authors explain why JNM-9-ABC has a ABC stacking directly instead of having AA stacking?

Author's response: For mesoporous systems, due to their high surface area, it is expected to have large interfacial energy, leading to a natural trend of minimizing their free energy by closing energetically unfavorable pores (ref. *Macromol. rapid comm.* **2004**, 25, 1487-1490). Therefore, in the field of COFs, the increases of the length of linker did not always produce the large pore size (See ref. *Chem. Sci.*, 2019, 10, 4293). Moreover, although the theoretical calculation can be used to reveal the more energetically favorable stacking models, the change of synthetic condition often alters the interlayer structure. For instance, Cao's group (See ref. *ACS Appl. Mater. Interfaces* 2021, 13, 29471–29481) has been synthesized Tp-BTD COF with three different stacking structure by alteration of synthetic method. Specifically, change the solvent from *n*-butyl alcohol to dioxane, the AA stacking structure will be obtained instead of AB stacking structure. Zhang's group (See ref. *Small* 2023, 2303684) have been prepared NKCOF-11-AA and NKCOF-11-ABC by alteration of synthetic method, although the total stacking energy calculation indicated that AA stacking modes (140.18 kcal mol⁻¹) was more energetically favorable than that of ABC stacking (83.04 kcal mol⁻¹). Overall, it is still very hard to precisely predict the interlayer stacking structure in 2D COFs, since the DFT calculation only can be conducted without consideration of solvent and catalysts.

Nevertheless, we also conducted the DFT calculation to evaluated the total stacking energy of **JNM-9**. As shown in Table R1, the total stacking energy of AA (105.81 kcal mol⁻¹) and ABC stacking (60.95 kcal mol⁻¹) was much higher than AB stacking (39.13 kcal mol⁻¹). Therefore, without consideration of solvent (i.e., DMF, *o*-DCB and *n*-BtOH) and catalysts (i.e., TFA), the AA stacking structure is the most energetically favorable. But, we have tried several synthetic conditions and AA stacking structure was not obtained.

Table R1. The space group, cell parameters, total energy, and total crystal stacking energy per layer of the possible structures of **JNM-9** under different states.

Structure	Space group	Formula	c (Å)	Total energy (kcal/mol)	Total crystal stacking energy per layer (kcal/mol) ^a
Monolayer				-3773315.589	
JNM-9-AA	P6	C ₉₆ H ₆₆ N ₁₈ Cu ₆	3.5215	-3773421.396	105.8065731
JNM-9-AB	P63/m	C ₁₉₂ H ₁₃₂ N ₃₆ Cu ₁₂	6.8003	-7546709.441	39.13155517
JNM-9-ABC	R3	C ₂₈₈ H ₁₉₈ N ₅₄ Cu ₁₈	10.2004	-11320129.62	60.95219402

^a $\Delta E_{AA}/\text{layer} = E_{\text{monolayer}} - E_{AA}$, $\Delta E_{AB}/\text{layer} = (E_{\text{monolayer}} \times 2 - E_{AB})/2$, $\Delta E_{ABC}/\text{layer} = (E_{\text{monolayer}} \times 3 - E_{ABC})/3$

3. Figure 2; the colors in Figure 2 are very difficult to distinguish. I advise to change the colors of these plots to make them clearer.

Author's action: We thank this reviewer for pointing out this issue, and we have changed the colors of these plots to make them easier distinguish in Figure 2.

4. Figure 3: HRTEM measurements are much more important than SEM and low-resolution TEM, but in Figure 3, the authors only present the TEM images with crystal lattice in insets, which is very confusing. Please provide HR TEM data and present the electron diffraction mode.

Author's response: We thank this reviewer's suggestion and we revised Figure 3 as shown here as Fig. R2. The high-resolution transmission electron microscopy (HR-TEM) and Fourier transform (FFT) of JNM-7-AA, JNM-8-AA, and JNM-9-ABC demonstrated that the well-ordered lattice fringe with d-spacing of 3.10, 3.70, and 3.10 nm, corresponding to the lattice planes of (100), (100) and (110), respectively (Figs. 3b, 3d, and 3f). This result is in good agreement with their refined PXRD pattern.

Fig. R2. SEM images of (a) JNM-7-AA, (c) JNM-8-AA, and (e) JNM-9-ABC. HR-TEM images of (b) JNM-7-AA, (d) JNM-8-AA, and (f) JNM-9-ABC. Top right: enlarged images of a selective area showing well-ordered lattice fringe. Bottom right: fast Fourier transform (FFT) pattern.

5. Figure 4: how do the authors rule out the possibility that the topology of COFs changed rather than the change of layer stacking?

Author's response: It is well-known that the honeycomb nets can be obtained by imine condensation between monomers with C₃ and C₂ symmetry (See *Chem. Rev.*, 2020, 120, 8814–8933). More importantly, the PXRD pattern of **JNMs** exhibited characteristic peaks that are similar to 2D honeycomb structure. Recently, a 3D COF have been prepared from 2D honeycomb COF through inclined interpenetration (see *J. Am. Chem. Soc.* 2023, 145, 13537), however, the PXRD of 3D COF featured much more complicated peaks compared to 2D COF. In addition, the BET analysis also revealed narrow pore size distribution of **JNMs** that similar to 2D honeycomb structure. Combining all these evidences, we believe it is the alteration of interlayer stacking rather than topology change.

6. The N₂ sorption capacity of these materials are extremely low for all COFs, which might be resulted from the low crystallinity of these materials. Please improve the quality of these COFs.

8. In Figure 4d, the difference of 2.26 and 1.2 is 1, and the difference of 2.95 and 2.26 is 0.7. Is it hard to say which is closer based on so minor difference. This is the same for JNM-8. Another question is that why does the pore size not go back to the original size? Is this because of partial transformation? If so, why are there no two peaks for the recovered samples? Please explain this.

Author's response: Comment 6 and 8 are highly related, therefore, we reply to these comments together. Thank you very much for your comments. The BET surface areas of porous materials are not only related to their crystallinity but also highly affected by the activation processes. In addition, in our work, due to the incorporation of Cu-CTU, the density of CuOF will be higher compared to corresponding COF with similar pore size. During the structure transformation, the heating and magnetic stirring are required, which will reduce the crystallinity and particle size of samples (Supplementary Fig. 24.) and might block the pore of **JNMs**. Therefore, the relatively low BET surface areas and smaller pore size distribution were observed.

To improve the quality of samples, we carefully washed the samples by soxhlet extraction with various solvent, and then, the sample were activated by supercritical carbon dioxide before the N₂ sorption measurements. According to this reviewer's suggestion, we have retired the N₂ sorption experiment and the results have been shown in the Fig. R3. It clearly showed that the BET surface areas of all CuOF are largely improved. More importantly, their pore size distributions are more reasonable. For instance, **JNM-7-AA** (pristine) and **JNM-7-AA** (regenerated) exhibited closer pore size distribution (3.4 and 3.0 nm), which is much more difference with the **JNM-7-ABC** with a pore size distribution of 1.4 nm. Same for **JNM-8**. After improve the quality of sample, the pore size of **JNM-7** and **JNM-8** can almost go back to the original size (3.4 and 3.0 nm for **JNM-7-AA** pristine and regenerated samples, and 4.0 and 3.9 nm for **JNM-8-AA** pristine and regenerated samples).

Fig. R3. BET surface analysis of (a) **JNM-7** and (c) **JNM-8**, and pore size distribution profiles of (b) **JNM-7** and (d) **JNM-8**.

Author's action: we have revised Fig. 4 in the manuscript and Supplementary Fig. 31 in SI.

7. Since the N₂ sorption is so low, is it accurate to determine the pore size distribution via such low sorption isotherms?

Author's response: Thank you very much for your comment. Taking the **JNM-7-ABC** as an example to explain how we determine the pore size distribution. **JNM-7-ABC** showed a Type-I microporous sorption isotherm, and the BET surface area was calculated to be 157.9 m² g⁻¹ by using the BEL-Master software (Fig. R4). To accurately determine the pore size distribution of **JNM-7-ABC**, we chose the nonlinear density functional theory (NLDFT) slit-pore model. As shown in Fig. R4, the experimental N₂ adsorption isotherm of **JNM-7-ABC** was in good agreement with the adsorption isotherms simulated by the NLDFT slit-pore model, demonstrating the reliability of the pore size distribution. The calculated pore size of **JNM-7-ABC** is centered at 1.4 nm, which was close to the simulated pore size (1.3 nm) for the eclipsed ABC stacking model.

Fig. R4. (a) N_2 adsorption and desorption isotherms of **JNM-7-ABC** at 77 K. (b) BET plot for surface area calculation. (c) The experimental and simulated N_2 adsorption isotherm of **JNM-7-ABC**. (d) Calculated pore-size distribution of **JNM-7-ABC** based on the adsorption branch of the N_2 isotherm at 77 K.

9. The DFT calculation is very unreasonable. The DMF molecule in AA-stacking model is so close to the skeleton, while the DMF molecule in ABC-stacking model is adjacent to the skeleton. It is very apparent that closer distance will definitely result in higher energy levels. Additionally, the calculation does not make too much sense as these results are based on theoretical models. All DFT researchers know that ABC stacking will lead to higher energy levels than AA stacking regardless the materials, so the conclusion is very obvious and not surprising.

Author's response: Due to the larger pore size in AA-stacking, after optimization, the DMF molecule is far away to the frameworks with a distance larger than 10 \AA , while it is close to the skeleton in ABC stacking model with distance ranged from 2.297 to 5.672 \AA . As the reviewer suggested, the closer distance will lead to higher energy levels. Therefore, the DFT calculation suggests that the energy input is necessary for the interlayer structural transformation process, which is consistent with the experimental observations. Specifically, **JNM-7-AA** in DMF **cannot** transfer to **JNM-7-ABC** without heating and stirring.

Notably, the AA stacking is **NOT** always more energetically favorable than ABC stacking. For instance, Cui and co-authors have found that *the total crystal stacking energy of ABC stacking ($116.52 \text{ kcal mol}^{-1}$) for **COF1-*i*Pr** is much higher than those of the AA ($44.67 \text{ kcal mol}^{-1}$) and AB ($70.33 \text{ kcal mol}^{-1}$) stacking, and the PXRD of **COF1-*i*Pr** confirmed it had ABC stacking model (See ref. *J. Am.**

Chem. Soc. 2018, 140, 16124–16133). Therefore, it should be beneficial to conducted the DFT calculation for further understanding the structural transformation.

10. SEM EDS is not as accurate as TEM. Since the authors can get TEM measurements, why not use TEM EDS for elemental mapping?

Author's action: We thank this reviewer's suggestion, and we retested EDS during TEM measurements as shown in Fig. R5-Fig. R7. These Figures have been added into SI as Supplementary Fig. 11-13.

Fig. R5. EDS of JNM-7-AA. (a) TEM photo of JNM-7-AA (scale bar, 250 nm); Element mapping of (b) Cu: Blue; (c) C: Red; (d) N: Green.

Fig. R6. EDS of JNM-8-AA. (a) TEM photo of JNM-8-AA (scale bar, 500 nm); Element mapping of (b) Cu: Blue; (c) C: Red; (d) N: Green.

Fig. R7. EDS of **JNM-9-ABC**. (a) TEM photo of **JNM-9-ABC** (scale bar, 500 nm); Element mapping of (b) Cu: Blue; (c) C: Red; (d) N: Green.

11. Please provide elemental analysis data for these COFs in this work.

Author's action: We have provided the elements analysis data and revised in manuscript and pages 4–5 of Supplementary.

12. What is the possible mechanism for the claimed layer stacking? Is there any proof for this mechanism? Please clarify.

Author's response: It is extremely difficult to experimental prove and reveal the interlayer shifting mechanism in the field of COFs, and so far, it is acceptable to demonstrate the interlayer shifting mechanism by combining DFT calculation and PXRD measurements among the community (See ref. *J. Am. Chem. Soc.* 2020, 142, 12995–13002; *J. Am. Chem. Soc.* 2022, 144, 20363–20371, *J. Am. Chem. Soc.* 2018, 140, 12922–12929; *J. Am. Chem. Soc.* 2018, 140, 16124–16133; *ACS Appl. Mater. Interfaces* 2021, 13, 29471–29481; *Small* 2023, 2303684).

Nevertheless, in our case, we assumed that the DMF molecules have disturbed the Cu-Cu interactions between layers, leading to the AA stacking transfer to ABC stacking. To further support our assumption, we have tried various solvent such as DMF, dimethylacetamide (DMA), H₂O, dioxane, and CH₃OH under same condition (i.e., 80 °C for 8 h). As shown in Fig. R8. We have found that DMF and DMA can trigger the structure transformation of **JNM-7** and **JNM-8** from AA to ABC stacking model. However, other solvent like dioxane, and CH₃OH cannot induce structure change. In addition, the immersion of **JNM-7** into H₂O only reduced the crystallinity, while can partially trigger the structure transformation of **JNM-8** from AA to ABC stacking model. It is well known DMF, and DMA have strong coordinating ability toward transition metals than alcohols and esters (See ref. *Chem. Eur. J.* 2020, 26, 4350 – 4377, *Dalton Trans.*, 2011, 40, 10742–10750). These results implied that the solvent

with strong coordinating ability might trigger the structure transformation. We further tried stronger coordinating solvent like 1,8-diazabicyclo(5.4.0)undec-7-ene (DBU). As shown in Fig. R9, the addition of DBU could increase the crystallinity of **JNM-7-ABC** after the structural transformation. These experimental results further supported that coordinating solvent can disturb the Cu-Cu interactions between layers, leading to the AA stacking transfer to ABC stacking.

Fig. R8. PXRD analysis of (a) **JNM-7-AA** and (b) **JNM-8-AA** after treatment with various solvent at 80 °C for 8 h.

For the transformation between ABC and AA triggered by TFA, we assumed that the imine linkages can be protonated by TFA, resulting in the charge repulsion between layers and reconstruction of interlayer structure. According to the DFT calculation results, the disturbed layers eventually will go to the energetically most favorable structure (e.g., AA stacking model) and the Cu-Cu interactions in AA stacking model will also compensate the charge repulsion between layers. When HOAc, a weaker acid, was used instead of TFA, the **JNM-7-ABC** cannot transfer to **JNM-7-AA**, further confirm the protonation of imine bond might be important for the structure transformation.

Fig. R9. PXRD analysis of **JNM-7-AA** after treatment with a mixture of (a) DMF and various base (i.e., DBU and ammonia (aq.)) and (b) DCB with acid (i.e., TFA and HOAc) at 80 °C for 8 h.

13. I noticed that this manuscript has many typos or grammar issues. Please check through very carefully and improve the readability of this work.

Author's action: We thank this reviewer's suggestion, and we have carefully checked and polished the English.

Reviewer 2 (Remarks to the Author):

The authors have prepared three 2D COFs composed from Cu-CTU and three di-amine linkers with different length. Interestingly, the interlayer stacking structure of JNM-7 and JNM-8 can be reversibly modulated, leading to the reversible encapsulation and release of lipase. Due to the alteration of stacking, AA stacking structure exhibited better photocatalytic activity for oxidative cross-coupling reaction. Obviously, it is very hard to achieve reversible structure transformation and obtain permanent AA and ABC structure with the identical components. Thus, these results are of great importance and deserve publication in such a reputed journal as Nat. Comm. Moreover, the work is solid, the materials are thoroughly characterized, and structural transformation mechanism is well investigated. I think it is a very good addition to COF and MOF area and am happy to see it published after addressing some points:
Author's response: We thank this reviewer for finding our manuscript worthy of publishing in *Nature Communications* and for raising some questions to improve our work.

1. The ABC stacking with $R3$ space group should be trigonal, not hexagonal.

Author's action: We thank this reviewer for pointing out this issue, and we have changed ABC tacking with hexagonal $R3$ space group to trigonal $R3$ space group.

2. For the reversible interlayer structure transformation, did author try other solvent instead of DMF?

Author's response: We thank this reviewer for pointing out this issue. Please see the response for **Reviewer 1's Comment 12**.

3. From the BET analysis, it seems the as synthesized JNM-7-AA have a smaller pore at 1.20 nm, did this can be attributed to ABC stacking model?

Author's response: We thank this reviewer for pointing out this issue, and we simulated the **JNM-7-ABC** structure and measured its pore size to be ~ 1.40 nm, which is similar to the experimentally measured pore size (~ 1.30 nm) after the **JNM-7-AA** layer-stacking structure conversion. This further demonstrates the success of the layer stacking transformation.

4. The For elemental analysis of JNMs, the calculated and observed ratio are not in good agreement with each other. It's better to re-check.

Author's action: We have retried elements analysis and provided in manuscript and SI.

Reviewer 3 (Remarks to the Author):

In this paper, the authors reported the synthesis of a series of CuOFs, and their reversible structural transformation triggered by acid or heat. Due to the modulation of interlayer stacking model, the porosity and photocatalytic performance can be tuned. The materials and properties were convincingly and systematically studied including crystal structure, porosity, stability, host-guest chemistry and catalytic performance. In addition, the authors extensively investigated the reversible structural transformation and its mechanism. This paper is interesting and would be a valuable contribution to the field. Therefore, this reviewer recommends its publication in Nat. Comm., after minor revisions after the following comments are addressed.

Author's response: We also thank this reviewer for supporting our manuscripts published on Nature Communication.

1. The space group "P6/M" should be changed to "P6/m".

Author's action: We thank this reviewer for pointing out this issue, and we have changed the space group "P6/M" to "P6/m in the manuscript and Supplementary.

2. Graphics and texts are inconsistent. In Figure 5a, it shows that LP@JNM-8-AA was heated in DMF to release the LP. But in manuscripts, "...after centrifugation, the collected powder of LP@ JNM-8-AA was added to phosphate buffer solution and the resulting mixture was heated at 80°C for 6 h. Lipase(Figs. 5b, and 5c)". Please check and revise.

Author's action: We thank this reviewer for pointing out this issue, and we have modified Figure 5a and "...the collected powder of LP@ JNM-8-AA was added to phosphate buffer solution and the resulting mixture was heated at 80°C for 6 h..." have changed to "...the collected powder of LP@ JNM-8-AA was added to phosphate buffer solution and DMF solution and the resulted mixture was heated at 80 °C for 6 h..." in the manuscript.

3. In the Supplementary Table 12, did the author calculate the TOF based on one Cu ion or one Cu-CTU (three Cu ions)?

Author's action: The TOF based on one Cu-CTU (three Cu ions) and have explained in Supplementary Table 12.

4. In the Supplementary Figure 35, it should be the signal of $^1\text{O}_2$ instead of O_2^- .

Author's action: We have changed O_2^- to $^1\text{O}_2$ in the Supplementary Figure 35.

5. The theoretical and experimental values of the elemental analysis of JNM-7-AA, JNM-8-AA and JNM-9-ABC are quite different.

Author's action: We have retried elements analysis and provided in manuscript and SI.

REVIEWERS' COMMENTS

Reviewer #1 (Remarks to the Author):

The authors have addressed most of my concerns even though I still cannot agree with some explanations. I do understand that chemical science has its limitations at this time stage and many things cannot be explained very well. I recommend it for publication after checking the manuscript writing.

Reviewer #2 (Remarks to the Author):

The authors have sufficiently addressed my remaining concerns. I now recommend the article for publication.

Reviewer #3 (Remarks to the Author):

The authors have satisfactorily addressed all the comments from the reviewer and the manuscript can now be accepted in its current form.

Comments from Reviewers and Our Responses

(Black lines: original comment from the reviewers; Blue lines: author's response)

Reviewer #1 (Remarks to the Author):

The authors have addressed most of my concerns even though I still cannot agree with some explanations. I do understand that chemical science has its limitations at this time stage and many things cannot be explained very well. I recommend it for publication after checking the manuscript writing.

Author's response: We are grateful to the reviewers for acknowledging our revision and supporting the publication in *Nature Communications*. English has been polished.

Reviewer #2 (Remarks to the Author):

The authors have sufficiently addressed my remaining concerns. I now recommend the article for publication.

Author's response: We thank the reviewer for careful evaluation of our manuscripts and acceptance of our efforts in the improvement of our work.

Reviewer #3 (Remarks to the Author):

The authors have satisfactorily addressed all the comments from the reviewer and the manuscript can now be accepted in its current form.

Author's response: We also thank this reviewer for finding our manuscript worthy of publishing in *Nature Communications*.